# Unraveling Macroplastic Pollution in Rural and Urban Beaches in Sarangani Bay Protected Seascape, Mindanao, Philippines

Frank T. Acot, Jr. [1], Recca E. Sajorne [2,*], Nur-Ayn K. Omar [3,4], Peter D. Suson [1], Lynn Esther E. Rallos [5] and Hernando P. Bacosa [1]

1. Environmental Science Graduate Program, Department of Biological Sciences, College of Science and Mathematics, Mindanao State University-Iligan Institute of Technology, Iligan 9200, Lanao del Norte, Philippines
2. College of Fisheries and Aquatic Sciences, Western Philippines University-Puerto Princesa Campus, Puerto Princesa 5300, Palawan, Philippines
3. College of Education, Mindanao State University-Main Campus, Marawi 9700, Lanao del Sur, Philippines
4. Main Campus Bataraza Extension, Mindanao State University, Bataraza 5306, Palawan, Philippines
5. Science Department, College of Natural Sciences and Mathematics, Mindanao State University-General Santos, Tambler 9500, General Santos, Philippines
* Correspondence: sajornerecca@gmail.com; Tel.: +63-956-904-5767

**Abstract:** Plastic pollution in the ocean is an emerging environmental concern in the Philippines. This study was conducted to investigate the prevalence of macroplastics, composition of plastic litter, and the clean-coast index (CCI) of urban and rural beaches in Sarangani Bay. Plastic litter was collected by delineating a 100-m transecting line with three 4 m × 4 m quadrats. The density of macroplastic litter in urban areas (0.66 items m$^{-2}$) was significantly higher than in rural areas (0.29 items m$^{-2}$). The plastics sampled were predominantly food packaging, such as polyethylene bags, which are locally known as *sando* bags. The accumulation rate of macroplastic litter ranged from 0.07 items d$^{-1}$ m$^{-2}$ to 0.40 items d$^{-1}$ m$^{-2}$, in which urban beaches (0.25 items d$^{-1}$ m$^{-2}$) have a significantly higher accumulation rate than rural beaches (0.11 items d$^{-1}$ m$^{-2}$). Overall, the calculated CCI of the beaches of Sarangani Bay was categorized as clean to moderately clean for rural beaches and moderately clean to extremely dirty for urban beaches.

**Keywords:** clean-coast index (CCI); marine litter; accumulation; density

## 1. Introduction

Plastic pollution in the ocean is a rapidly emerging global environmental concern, with high concentrations (up to 580,000 pieces per km$^2$) and a global distribution, driven by exponentially increasing production [1]. The exponential increase in the use of plastics in modern society and the inadequate management of the resulting waste have led to plastic accumulation in the marine environment [2]. This is a major environmental issue that affects coasts and coastlines all around the world [3].

Plastics are widely used due to advantageous properties, such as elasticity, hardness, lightness, transparency, and durability. As a result, the scale of plastic production has risen dramatically from an annual volume of 0.5 million metric tons in the 1940s to 550 million tons in 2018 [4]. Based on size, plastic debris is usually categorized as megaplastics (>1 m), macroplastics (2.5 cm–1 m), mesoplastics (5 mm–2.5 cm), microplastics (1 μm–5 mm), and nanoplastics (<1 μm) [5,6]. Microplastics (MP) can be categorized as primary MP, which is manufactured at that size, and secondary MP, which is a result of the breakdown of macroplastics due to fragmentation and degradation [7].

According to a model developed by Meijer et al. [8], the Philippines is the largest emitter of plastics from rivers to the marine environment. The country also ranks third in total plastic waste input from land into the ocean, just below China and Indonesia, by

contributing 0.28–0.75 million metric tons of plastic litter every year [9]. These findings are mainly based on simulation and numerical modeling, empirical observations, and a limited amount of field data. Validating the results of these findings through a more representative field sampling is crucial to science-based decision-making. Several studies on plastic pollution monitoring were conducted on the beaches of the Philippines, and the majority of the beaches were categorized as dirty to extremely dirty based on the clean-coast index (CCI) [10–12]. These beach areas were mostly categorized as urban areas where the rate of infrastructure development is higher than in rural areas. Aside from this, activities such as fishing and swimming are also frequently conducted by the local community and beachgoers, which helps development and economic growth, particularly in local coastal communities [13]. However, these activities may contribute to plastic pollution as indiscriminate disposal of solid waste is highly rampant in coastal settlements [12,14–16]. This emerging threat requires the development and enforcement of plastic regulation policies. In a study conducted by Galarpe et al. [17], it was determined whether studies on macro- and microplastics should be the basis of the institutionalization of existing policies on waste management. Unfortunately, they were unable to arrive at sound conclusions due to a limited number of studies on macro- and mesoplastics in the Philippines, and therefore a limited amount of field data, considering that the country is seen as a plastic pollution hotspot around the world.

The intensity of human pressure on marine systems has led to a push for stronger marine conservation efforts. The Philippines has been the site of some of the earliest marine reserves and marine-protected areas [18]. Marine reserves, as defined by [19], are areas of marine environment protected from various forms of human exploitation. Within the last decade, South Cotabato-Sarangani-General Santos City (SOCSARGEN) in the southern Philippines has emerged as one of the most economically dynamic regions in the Philippines [20]. A significant factor propelling growth in the area is the presence of Sarangani Bay Protected Seascape (SBPS), a marine-protected area that provides various goods and services that cater to the SOCSARGEN economy. The bay encloses an area of 449.22 km$^2$ and is bounded by the Sarangani Province and chartered city of General Santos. Besides providing a sanctuary for marine life, the bay also offers a wide range of choices for recreational activities due to the presence of white-sanded beaches, beautiful coral reefs, and scuba diving sites [21]. Despite its status as a marine-protected area, the bay has faced several environmental pressures, including massive population growth, uncontrolled human settlement in coastal areas, urbanization, and various anthropogenic pressures, including pollution [22]. However, the plastic pollution in the area has not been documented. Thus, with the growing concern of plastic pollution in the Philippines, this study was conducted to (1) determine the density of macroplastic litter on the rural and urban beaches, (2) identify the types of macroplastic litter, (3) quantify the accumulation rate of macroplastic litter for eight non-consecutive days, and (4) determine the clean-coast index of rural and urban beaches of Sarangani Bay Protected Seascape. To the best of our knowledge, this is the first monitoring of marine plastic litter pollution in the protected seascape of the Philippines.

## 2. Materials and Methods

The study was conducted in Sarangani Bay Protected Seascape (Figure 1). Sarangani Bay is located on the southern tip of Mindanao Island in the Philippines. It opens up to the Celebes Sea in the Pacific Ocean. The bay was declared a protected seascape in 1996 (Proclamation no. 756 signed by President Fidel V. Ramos), covering an area of 215,950 hectares. Plastic litter was sampled from 14 beaches in rural (n = 7) and urban (n = 7) areas in Sarangani Bay, and the sampling sites were classified as urban (U) or rural (R) (Table 1).

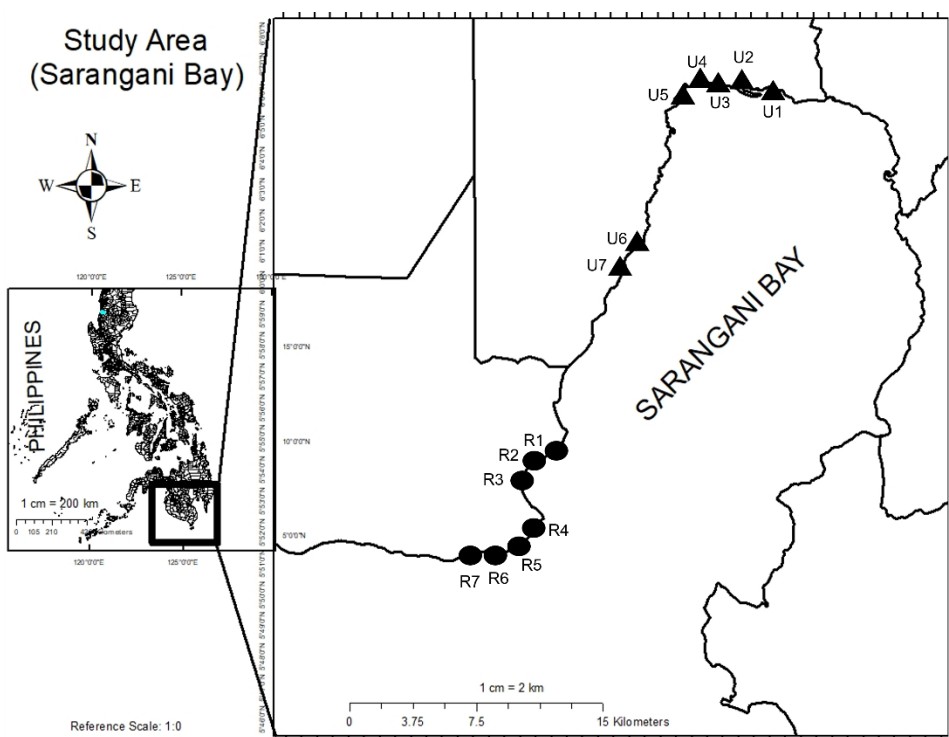

**Figure 1.** The map of Sarangani Bay bounded by Sarangani Province and the chartered city of General Santos showing the sampling sites in urban (triangle) and rural (circle) areas.

**Table 1.** Description of the study sites and macroplastics sampled from the beaches of Sarangani Bay, Philippines; SD—represents standard deviation of the replicates.

| Site Code | Latitude | Longitude | Classification | Total Litter | Density (Items m$^{-2}$) $\pm$ SD |
|---|---|---|---|---|---|
| U1 | 6°05′57.5″ N | 125°12′01.7″ E | Urban | 216 | 0.56 $\pm$ 0.22 |
| U2 | 6°05′58.4″ N | 125°11′52.3″ E | Urban | 251 | 0.65 $\pm$ 0.51 |
| U3 | 6°06′12.1″ N | 125°11′30.6″ E | Urban | 410 | 1.06 $\pm$ 0.74 |
| U4 | 6°06′22.1″ N | 125°10′41.9″ E | Urban | 339 | 0.88 $\pm$ 0.33 |
| U5 | 6°06′23.4″ N | 125°10′28.7″ E | Urban | 317 | 0.82 $\pm$ 0.16 |
| U6 | 6°02′22.5″ N | 125°08′36.7″ E | Urban | 116 | 0.30 $\pm$ 0.11 |
| U7 | 6°02′16.0″ N | 125°08′34.9″ E | Urban | 139 | 0.36 $\pm$ 0.13 |
| R1 | 5°54′14.4″ N | 125°04′55.0″ E | Rural | 180 | 0.46 $\pm$ 0.23 |
| R2 | 5°54′07.4″ N | 125°04′48.3″ E | Rural | 165 | 0.42 $\pm$ 0.16 |
| R3 | 5°54′02.0″ N | 125°04′45.0″ E | Rural | 131 | 0.34 $\pm$ 0.14 |
| R4 | 5°52′18.6″ N | 125°05′05.5″ E | Rural | 89 | 0.23 $\pm$ 0.19 |
| R5 | 5°52′08.9″ N | 125°04′58.0″ E | Rural | 71 | 0.18 $\pm$ 0.11 |
| R6 | 5°51′49.3″ N | 125°04′45.8″ E | Rural | 72 | 0.19 $\pm$ 0.14 |
| R7 | 5°51′41.8″ N | 125°04′37.7″ E | Rural | 84 | 0.21 $\pm$ 0.18 |

Sampling was performed by following the methods of Sajorne et al. [12] with minor modifications. At each sampling site, a 100-m transecting line was established parallel to the shoreline. Three quadrats measuring 4 m × 4 m were laid along the transecting line. Plastic litter found within the quadrat was collected, counted, and classified. The collected plastics were classified based on their uses, such as food packaging, polyethylene bags (locally known as *sando* bags), Styrofoam, mats, plastic bottle, plastic fragments, plastic caps, toiletries, Styrofoam cups, clothing, and other items (Figure S1) [12,23]. Meanwhile, to monitor the accumulation rate of macroplastics, non-consecutive sampling was conducted for eight days in beach areas [16]. The transecting line and quadrats were placed in the same area in each sampling site throughout the monitoring. Sampling collection was conducted

from 11 August 2021 until 4 September 2021. The collection of macroplastics was conducted on Wednesdays to represent the weekdays and Saturdays to represent the weekends

The density of plastic litter was determined by dividing the total number of items per unit area sampled [24,25] and reported as items/m$^2$ ± SD. The cleanliness of the coastal beaches was assessed using the clean-coast index (CCI) [26], where CCI is equal to the density of litter items/m$^2$ multiplied by 20, a constant to make the numerical value of CCI more intuitive. According to the CCI scale, values from 0 to 2 are very clean, 2 to 5 are clean, 5 to 10 are moderately clean, 10 to 20 are dirty, and greater than 20 are extremely dirty. Meanwhile, the daily accumulation rate, mean count, and cumulative density were computed using the formula of Ammendolia et al. [27]. The accumulation rate was reported as items d$^{-1}$ m$^{-2}$ [28].

An independent two-sample *t*-test was conducted to determine significant differences between urban and rural beaches in Sarangani Bay in terms of macroplastic abundance as the data are parametric. Statistical analysis was performed in Statistical Package for the Social Sciences (SPSS) version 26, a free software, developed by International Business Machines Corporation (IBM) headquartered in Armonk, New York, and values of $p < 0.05$ were considered significantly different.

## 3. Results

Regarding the sampling sites, all beaches were found to contain macroplastic litter. A total of 2580 items of plastic litter were collected over a cumulative area of 672 m$^2$ in urban and rural areas of Sarangani Bay Protected Seascape, with a density of 3.83 items m$^{-2}$. The highest density was recorded in urban areas U3 (1.06 items m$^{-2}$), U4 (0.88 items m$^{-2}$), and U5 (0.82 items m$^{-2}$) (Table 1). Overall, the plastic litter density on urban beaches (0.66 items m$^{-2}$) was significantly higher ($p < 0.05$) than the plastic density on rural beaches (0.29 items m$^{-2}$) (Figure 2).

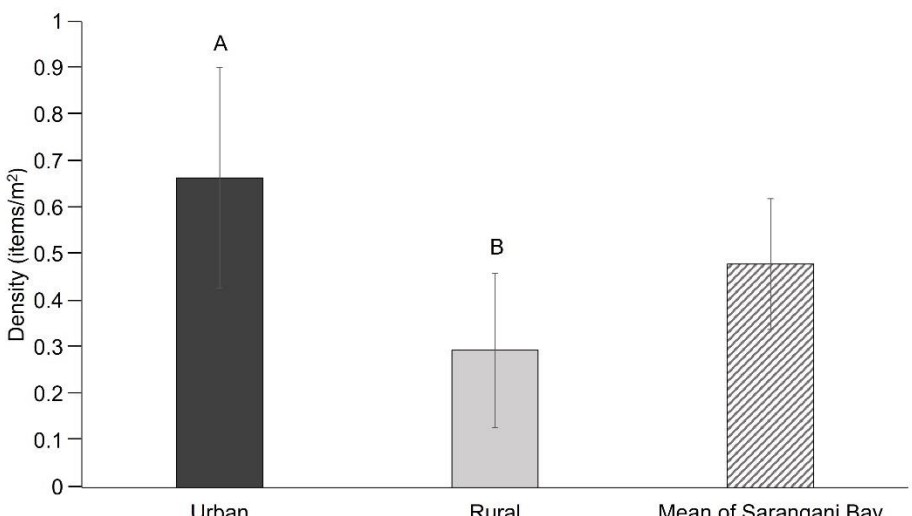

**Figure 2.** Density of macroplastics sampled from seven rural beaches and seven urban beaches of Sarangani Bay, Philippines. Error bars represent the standard deviation of 14 sampling times. Different letters indicate significant differences ($p < 0.05$).

In terms of macroplastic types, there is a difference in the percentage composition in rural and urban areas of Sarangani Bay Protected Seascape (Figure 3). In urban areas, food packaging (27%, n = 474 items), polyethylene bags or *sando* bags (15%, n = 271 items), and mats (13%, n = 237 items) were the most predominant types of macroplastic. Meanwhile, *sando* bags (23%, n = 180 items), food packaging (19%, n = 154 items), and Styrofoam (14%, n = 108 items) were predominant in the rural areas. Overall, the types of plastics sampled were predominantly food packaging (24%, n = 628), *sando* bags (18%, n = 451), and other

items (19%, n = 496). The "other items" were composed of combined plastic classifications that made up less than two (2%) percent of the total plastic types.

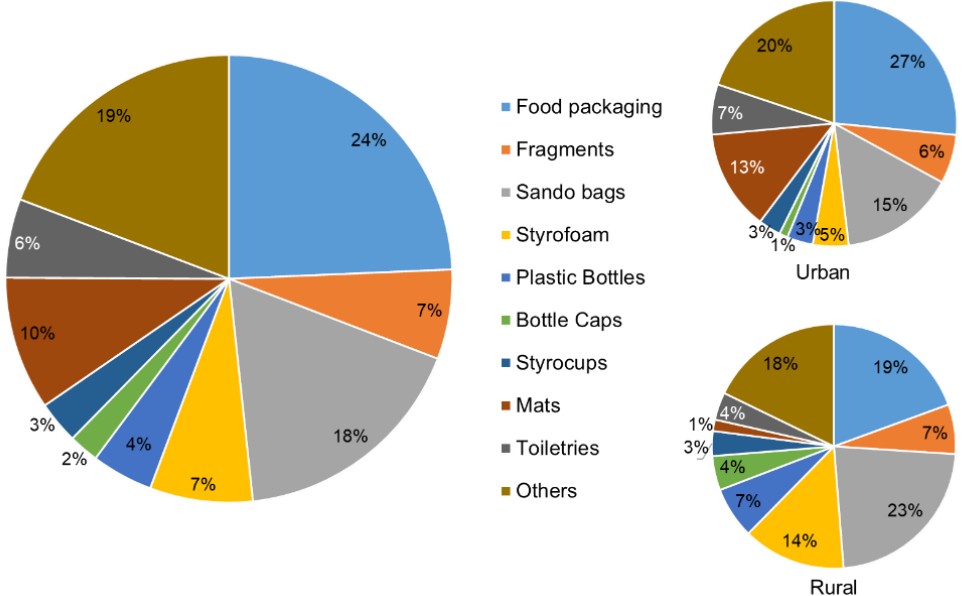

**Figure 3.** Types of macroplastic litter in rural and urban beaches in Sarangani Bay, Philippines.

Meanwhile, Sarangani Bay Protected Seascape was monitored for eight non-consecutive days, and it revealed that the daily density of macroplastic ranged from 0 to 2.52 items m$^{-2}$, with a mean density of 0.48 items m$^{-2}$ (SD = 2.26) (Table 2). The accumulation rates of macroplastic litter ranged from 0.07 items d$^{-1}$ m$^{-2}$ to 0.40 items m$^{-2}$, with an average rate of 0.18 items d$^{-1}$ m$^{-2}$. On average, the accumulation rate in urban areas (0.25 items d$^{-1}$ m$^{-2}$) was significantly higher than those in rural areas (0.11 items d$^{-1}$ m$^{-2}$) ($p$ = 0.007). Meanwhile, it was shown that day 2 of the collection was found to have the highest density (1.35 items d$^{-1}$) (n = 456), while the lowest density (0.75 items d$^{-1}$) (n = 255) was recorded on day 8 (Figure 4). Over the sampling period, a significant decrease in the density of plastic litter on both urban and rural beaches was not observed, which was characterized by a recurring amount or lingering presence of these pollutants in the area (Figure 4).

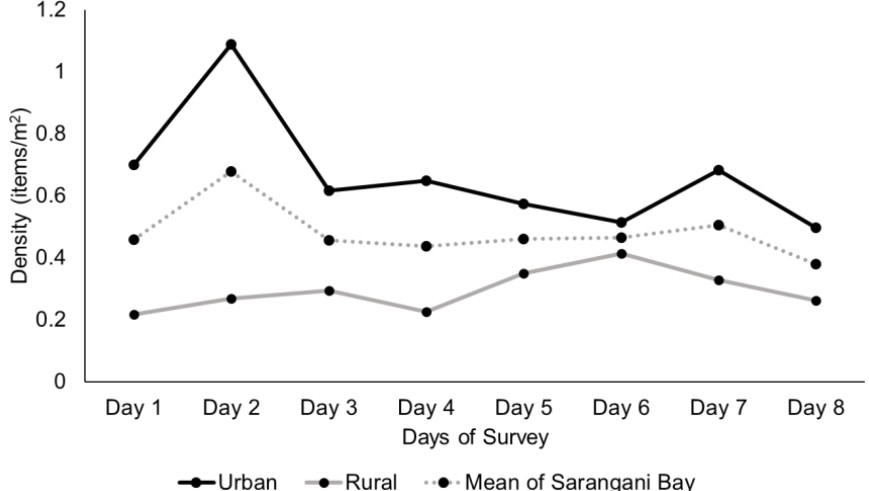

**Figure 4.** Temporal pattern of total plastic items collected in urban and rural beach areas for eight non-consecutive days.

**Table 2.** Summary of macroplastic surveys including the item counts, densities (items m$^{-2}$), and accumulation rates (items d$^{-1}$ m$^{-2}$) in sampling sites of Sarangani Bay Protected Seascape. The standard deviation (SD), daily minimum (min.), and maximum (max.) counts and densities are also presented.

| Segment | Area Surveyed (m²) | Number of Days Sampled (n) | Survey Frequency (Number of Surveys per Week) (n) | First-Day Count (n) | Total Count (n) | Daily Min. Count (n) | Daily Max. Count (n) | Daily Mean Count (n) | Daily SD Count (n) | Cumulative Density (Items m⁻²) | Daily Min. Density (Items m⁻²) | Daily Max. Density (Items m⁻²) | Daily Mean Density (Items m⁻²) | Daily SD Density (Items m⁻²) | Accumulation Rate (Items d⁻¹ m⁻²) |
|---|---|---|---|---|---|---|---|---|---|---|---|---|---|---|---|
| U1 | 48 | 8 | 2 | 24 | 216 | 14 | 43 | 27.00 | 10.61 | 4.50 | 0.29 | 0.90 | 0.56 | 0.22 | 0.21 |
| U2 | 48 | 8 | 2 | 28 | 251 | 11 | 82 | 31.38 | 24.72 | 5.23 | 0.23 | 1.71 | 0.65 | 0.52 | 0.25 |
| U3 | 48 | 8 | 2 | 35 | 410 | 15 | 121 | 51.25 | 35.86 | 8.54 | 0.31 | 2.52 | 1.07 | 0.75 | 0.40 |
| U4 | 48 | 8 | 2 | 73 | 339 | 22 | 53 | 42.38 | 16.05 | 7.06 | 0.46 | 1.10 | 0.88 | 0.33 | 0.33 |
| U5 | 48 | 8 | 2 | 32 | 317 | 27 | 50 | 39.63 | 7.78 | 6.60 | 0.56 | 1.04 | 0.83 | 0.16 | 0.31 |
| U6 | 48 | 8 | 2 | 22 | 116 | 7 | 22 | 14.50 | 5.42 | 2.42 | 0.15 | 0.46 | 0.30 | 0.11 | 0.11 |
| U7 | 48 | 8 | 2 | 21 | 139 | 10 | 29 | 17.38 | 6.28 | 2.90 | 0.21 | 0.60 | 0.36 | 0.13 | 0.14 |
| R1 | 48 | 8 | 2 | 17 | 180 | 6 | 37 | 22.50 | 11.35 | 3.75 | 0.13 | 0.77 | 0.47 | 0.24 | 0.18 |
| R2 | 48 | 8 | 2 | 12 | 165 | 12 | 31 | 20.63 | 7.95 | 3.44 | 0.25 | 0.65 | 0.43 | 0.17 | 0.16 |
| R3 | 48 | 8 | 2 | 10 | 131 | 6 | 28 | 16.38 | 6.74 | 2.73 | 0.13 | 0.58 | 0.34 | 0.14 | 0.13 |
| R4 | 48 | 8 | 2 | 14 | 89 | 1 | 27 | 11.13 | 9.28 | 1.85 | 0.02 | 0.56 | 0.23 | 0.19 | 0.09 |
| R5 | 48 | 8 | 2 | 10 | 71 | 4 | 19 | 8.88 | 5.72 | 1.48 | 0.08 | 0.40 | 0.18 | 0.12 | 0.07 |
| R6 | 48 | 8 | 2 | 4 | 72 | 0 | 20 | 9.00 | 6.99 | 1.50 | 0.00 | 0.42 | 0.19 | 0.15 | 0.07 |
| R7 | 48 | 8 | 2 | 6 | 84 | 0 | 16 | 10.50 | 9.02 | 1.75 | 0.00 | 0.33 | 0.22 | 0.19 | 0.08 |
| Total | 673 | 8 | 2 | 73 | 2580 | 0 | 121 | 322.50 | 108.30 | 3.83 | 0.00 | 0.18 | 0.48 | 2.26 | 0.18 |

Based on the CCI of the beaches in Sarangani Bay Protected Seascape, 36% (5 out of 14) were considered dirty to extremely dirty (Table 3). Out of the seven urban beaches, only one (site U3) (21.35) was categorized as extremely dirty, with a high CCI. Four of the sampling sites (U1, U2, U4, and U5) (11.25 to 17.65) were categorized as dirty. However, there were still some beaches (U6 and U7) categorized as clean (Table 3). Meanwhile, on rural beaches, all of the sampling sites were considered as clean to moderately clean. Notably, dirty to extremely dirty sites were located in urban areas, while clean to moderately beaches were located in rural areas. Overall, on average, Sarangani Bay (9.59) is categorized as moderately clean.

**Table 3.** Characteristics of the beaches in urban and rural areas in Sarangani Bay based on clean-coast index (CCI) compared to other studies in the Philippines.

| Area | Site Code | CCI Score | CCI Description | Author |
|---|---|---|---|---|
| Bula | U1 | 11.25 | Dirty | This study |
| Bula | U2 | 13.07 | Dirty | This study |
| Bula | U3 | 21.35 | Extremely Dirty | This study |
| Dadiangas South | U4 | 17.65 | Dirty | This study |
| Dadiangas South | U5 | 16.51 | Dirty | This study |
| Banualan, Tambler | U6 | 6.04 | Moderately Clean | This study |
| Banualan, Tambler | U7 | 7.23 | Moderately Clean | This study |
| Tinoto, Maasim | R1 | 9.37 | Moderately Clean | This study |
| Tinoto, Maasim | R2 | 8.59 | Moderately Clean | This study |
| Tinoto, Maasim | R3 | 6.82 | Moderately Clean | This study |
| Kamanga, Maasim | R4 | 4.63 | Clean | This study |
| Kamanga, Maasim | R5 | 3.69 | Clean | This study |
| Kamanga, Maasim | R6 | 3.75 | Clean | This study |
| Kamanga, Maasim | R7 | 4.37 | Clean | This study |
| Binduyan, Puerto Princesa City | E1, E2 | 48.75, 32.08 | Extremely dirty | [12] |
| Lucbuan | E3, E4 | 47.5, 17.92 | Extremely dirty | [12] |
| San Manuel | E5 | 16.67 | Dirty | [12] |
| San Miguel | E6 | 45.83 | Extremely dirty | [12] |
| Bancao | E7 | 77.92 | Extremely dirty | [12] |
| Mangingisda | E8 | 17.08 | Dirty | [12] |
| Inagawan | E9, E10 | 111.25, 33.33 | Extremely dirty | [12] |
| Inagawan | E11 | 16.67 | Dirty | [12] |
| Cabayugan | W1 | 12.08 | Dirty | [12] |
| Buenavista | W2, W3 | 55.83, 32.92 | Extremely dirty | [12] |
| Bacungan | W4 | 0 | Very clean | [12] |
| Bacungan | W5 | 5 | Clean | [12] |
| Simpocan | W6 | 0 | Very clean | [12] |
| Simpocan | W7 | 22.08 | Extremely dirty | [12] |
| Napsan | W8,W10 | 0, 0 | Very clean | [12] |
| Napsan | W9 | 15.42 | Dirty | [12] |
| Bulua, CDO | | 33.33 | Extremely dirty | [11] |
| Bonbon, CDO, Macabalan | | 93.33 | Extremely dirty | [11] |
| Baloy, CDO | | 120 | Extremely dirty | [11] |
| Talim Bay | | 13.14 | Dirty | [10] |
| Opol | | 2.67 | Clean | [29] |
| El Salvador, Alubijid | | 2.27 | Clean | [29] |

## 4. Discussion

With these results, we confirmed the significant difference between the amount of macroplastic litter in rural and urban beaches in Sarangani Bay Protected Seascape. Urban beaches were noted to have two times higher macroplastic litter density compared to rural beaches. We observed that within the urban beaches, fishing activity was frequently conducted by the local community, while recreational activities were very seldom. However, no fishing paraphernalia was collected throughout the study period. Additionally, there

are several residential areas within the vicinity of the urban beaches which can be one of the main sources of plastic litter. It is believed that the number of people in an area has a significant relationship to the quantity of plastic waste disposal [12,13]. These factors might be the reason why we recorded the highest density and accumulation rate of macroplastic litter at urban beach sites U3 and U4. This result also supports the findings of Sajorne et al. [12,24], where the majority of the plastic litter was collected from residential and fishing areas. Meanwhile, several studies on plastic monitoring on the Philippine coasts also found plastic food wrappers/containers, cups, and sachet wrappers [10,11,29] as the dominant macroplastic waste. Basic necessities and commonly used plastic types are believed to be the underlying reasons for such findings. For instance, *sando* bags were listed as the dominant type of macroplastics collected in rural areas, while they were placed second in urban areas. This may be due to the need for *sando* bags for carrying food and other purchased items in rural areas, which may not be the same in urban areas where food and other items are in closer proximity to buyers. Living conditions and activities in urban areas seem to allow for easy purchases anytime and anywhere. In rural areas, however, stores are located far apart, which probably drives residents to purchase items in bulk, resulting in the use of more *sando* bags. During sampling, it was also observed that mats and clothing were littering the beaches of urban areas but were hardly seen in rural areas. In urban areas, residents can conveniently replace such items because of greater access to them, whereas rural residents are known to recycle both mats and clothing. Conversely, on rural beaches, activities related to tourism, such as swimming, were more frequently conducted by beachgoers than fishing activities. The high number of plastic cups in rural areas may be attributed to tourism as beachgoers or tourists usually bring disposables for special occasions or recreational activities [30]. This shows that the overall plastic usage in urban and rural areas varies. It can be said that the plastic composition is due to the necessities and activities of a particular place. The findings of this study, however, do not deviate from the findings of other studies that found plastic food wrappers and containers/*sando* bags to be the dominant plastic waste on or near the beaches of the Philippines [31].

The temporal pattern observed in this study reveals that plastic spells a big problem for the future as it is non-biodegradable. In a study conducted by Browne et al. [32], the influence of physical factors, such as wind, wave action, and the density of plastic litter, on the spatial pattern of the accumulation of plastic litter was highlighted. During the collection, it was observed that the difference in waste generation was due to geographical location, and the number of residents [33]. Rural areas were a bit more mountainous with rocky beaches, and the number of households was not yet populous, unlike urban areas where beaches were plainly sandy and the coastline was occupied with houses. Additionally, the rural areas were close to the open sea, while the urban areas were by the bay. These data suggest that the accumulation rate of plastic in a given area does not depend only on human population size, but also on the geographical formation and activity of the site. Other than that, we also consider the influence of winds and waves [34,35] and meteorological conditions [36] on the transport and accumulation of macroplastic litter in these areas. In comparison to other areas in the Philippines, the beaches of Sarangani Bay Protected Seascape are cleaner than the beaches of Puerto Princesa, Palawan (29) [12], and Macajalar Bay, Misamis Oriental (84.99) [11]. In this study, the urban sites of Sarangani Bay were found to be moderately clean, dirty, and extremely dirty. Our CCI findings agree with the findings of Esquinas et al. [11] and Kalnasa et al. [29], who found the urban sites of Baloy, Macabalan, Bonbon, and Balua as clean and extremely dirty. As aptly stated, the highly urbanized nature of the adjacent environment greatly influenced the litter deposition in the surface sand of Macajalar Bay [11].

This study also reveals that plastic litter is now recognized as an emerging threat to Sarangani Bay. Marine macroplastic litter not only negatively affects ecology and biodiversity [37] but also has a direct impact on the economic value [7] of the coastal community as Sarangani Bay is a protected seascape and uplifts the economy of the

SOCCSARGEN region through tourism. However, this study also revealed that tourism-related items were the dominant macroplastic litter found on the rural and urban beaches of Sarangani Bay. With the presence of plastic litter on coastal beaches, the beaches lose their aesthetic value [38], thus making them unattractive and unsuitable for recreational activities [39,40]. The advancement toward meeting the main tenets of the UN's Sustainable Development Goals (SDG) has been hampered by insufficient initiatives for solid waste management policies at regional and national levels.

Meanwhile, SDG 6—clean water and sanitation—and SDG 14—life below the water—are both directly impacted by the pollution brought on by incorrect solid waste disposal in coastal regions. Recent reports in Philippine marine waters have also revealed the first case of plastic ingestion in a rare Deraniyagala's beaked whale (*Mesoplodon hotaula*) [41] and green sea turtle (*Chelonia mydas)* [42]. As a result, there may be a loss of biodiversity in both aquatic and terrestrial ecosystems as well as a reduction in the livelihoods of the local coastal community [43]. Furthermore, if this continues, through the influence of physical and chemical factors [44], secondary MP and nanoplastics will eventually form and put the health of the local coastal community at risk [7]. Due to its high dependence on marine foods, SDG 3—good health and welfare of the local community—and SDG 12—responsible consumption and production—become vulnerable as a result of these records of plastic ingestion in biota. Pieces of evidence on microplastic ingestion in fish and other marine animals [45] have also been reported in the Philippines. Overall, the occurrence of marine macroplastic pollution in the urban and rural beaches of Sarangani Bay can be partially ascribed to inadequate regional and national initiatives for solid waste management in Mindanao.

## 5. Conclusions

This study is the first to provide an account of the current status of plastic litter on urban and rural beaches in Mindanao, Philippines. We observed that plastic litter is more prevalent in urban areas than in rural areas in terms of density. The collection patterns of plastic litter suggest that the number of plastics in a given area may either increase or decrease but is never zero, which means that plastic waste is always being deposited in coastal areas, possibly aided by wind and wave actions. Similar to other studies, the dominant types of plastic litter are food packaging and *sando* bags. The values of the clean-coast index in this study revealed that beaches in urban areas are moderately clean, dirty, and extremely dirty, while rural areas are clean and moderately clean. Since the urban and rural beaches of Sarangani Bay were classified over a range of moderately clean to extremely dirty, this implies that both beach areas require a stricter plastic waste policy implementation—one for maintaining the clean status and the other for rehabilitating dirty beaches. Furthermore, data on the identification of dominant macroplastic items can serve as a guide for legislators in creating policies to strengthen the promotion of environmentally friendly products to curb or limit the rapid production of these dominant items.

**Supplementary Materials:** The following are available online at https://www.mdpi.com/article/10.3390/jmse10101532/s1, Figure S1: Types of macroplastic litter based on their uses: (A) food packaging; (B) fragment; (C) polyethylene bag or *sando* bag; (D) Styrofoam; (E) plastic bottle; (F) bottle cap; (G) mat; (H) toiletry; (I) disposable cup.

**Author Contributions:** Conceptualization, F.T.A.J. and H.P.B.; investigation, F.T.A.J.; formal analysis, F.T.A.J., R.E.S., N.-A.K.O., P.D.S., L.E.E.R. and H.P.B.; writing original draft preparation, F.T.A.J., R.E.S. and N.-A.K.O.; writing—review and editing, P.D.S., L.E.E.R. and H.P.B.; supervision, P.D.S., L.E.E.R. and H.P.B. All authors have read and agreed to the published version of the manuscript.

**Funding:** This research received no external funding.

**Institutional Review Board Statement:** Not applicable.

**Informed Consent Statement:** Not applicable.

**Acknowledgments:** This study was supported by the Philippines' Department of Science and Technology through the Accelerated Science and Technology Human Resource Development Program (DOST-ASTHRDP). The first author would like to thank the DOST-ASTHRDP for the scholarship to pursue graduate studies at MSU-IIT and the Department of Environment and Natural Resources (DENR) Region 12, especially the Protected Area Management Board of Sarangani Bay Protected Seascape for allowing the researchers to conduct this study in Sarangani Bay.

**Conflicts of Interest:** The authors declare no conflict of interest.

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
