# Peer review of "Unraveling Macroplastic Pollution in Rural and Urban Beaches in Sarangani Bay Protected Seascape, Mindanao, Philippines"

_jmse, doi:10.3390/jmse10101532_

Round 1
Reviewer 1 Report
The manuscript needs to improve in several areas. At the moment it is a data reporting (report). However, the authors need to compare and analyze the information further. There are some comments on the content provided below.
A. How the first-day count was done? Could you please provide the first day (August 11) counts at each location?
B. Figure 1. The map is not clear. Please improve the quality of the map.
C. Table 2: Information is not clear. Some units are given in the first row of the table while some are given in the table title. Better providing all within the table.
D. Accumulation rate: Is it items/day? What is the use of this information?, this needs to be a common unit (As I understood the unit is items/day/16m^2, isn't it?). If so, think about providing the information in the common units such as items/day/m^2
E. Table 3- Authors provide locations Ws and Es, better indicating those in a map.
F. Conclusions should be specific to this study. Pelase avoids using common discussion points in the conclusion.
I.e. this has no relevance in the conclusions section -"Plastic litter in the ocean has been unveiled and is now recognized as a threat on a 247 global scale. Looking at its properties of buoyancy and durability, plastic floats on the sea 248 surface and can be transported over large distances and deposited into different marine 249 ecosystems. "
Reviewer 2 Report
The paper by Acot et al. analyzes plastic pollution on rural and urban beaches in the Philippines. I enjoyed reading the paper, the English is very good and the structure is clear. However, I have some comments that you can find below. I think that the manuscript would benefit from a more detailed discussion on potential macroplastic sources (such as fishery) and from a discussion on environmental factors (winds, waves) that might have influenced plastic accumulation. Also, please include photos of the plastic items that you found.
· L. 24: please, explain what a ‘sando bag’ is (you can also include a photo of a sando bag)
· Table 1: why is U3 especially polluted?
· Lines 126 – 137 should be moved to the Results.
· Figure 2: please, indicate that 7 rural and 7 urban beaches were sampled. There are no letters in the figure indicating significant differences.
· L. 153: please, provide photos of the mats and sando bags so that the readers can better understand what is meant here
· L. 165: it should be 'macroplastic' instead of 'microplastic'
· L. 170 and 171: could the macroplastic densities on these days have been influenced by environmental factors such as winds or waves?
· L. 190: what is PPE?
· Table 2: what could be the reason for the high accumulation rates at sites U3 and U4?
· L. 202: did you find macroplastics related to fishery?
· L. 217: do you really mean ‘caps’ here? Why should there be more caps in rural locations? Please, explain.
Reviewer 3 Report
Re: Unravelling Macroplastic Pollution in Rural and Urban 2 Beaches in Sarangani Bay Protected Seascape, Mindanao, Philippines
Abstract
Line 24: Change ‘plastic’ to ‘plastics’
Introduction
Ø Lines 42-43: Authors are advised to verify this categorization of plastics based on size, as it is against the norm. It is widely reported in the literature that microplastics range from 1µm to 5,000 µm (i.e. 0.001-5 mm) while nanoplastics are anything less than 1 µm (i.e. < 1000 nanometer). Plastic particles of 100, 200, 300, 500 µm have always been reported as microplastics, and they are all less than 1 mm. It is important to correct this size categorization in lines 42-43 and avoid a misrepresentation of facts.
Ø Lines 43-45: No reference? Besides, it is not absolutely correct to attribute the variation in plastic sizes to the reason(s) stated. Some (micro)plastics are primarily microplastics while some as a result of disintegration.
Ø Lines 57-60: Kindly rephrase the sentence.
Ø Lines 60-62: I see no relevance in this text.
Materials and methods
Ø Lines 104-113: Some clarifications are needed here. The authors mentioned that the accumulation rate of plastics was done for eight consecutive days. How was this done? Also, where were the quadrats laid on sampling days? It’s very important to know if quadrats were set at the same place or not since we are talking about the accumulation rate. Finally, can you shed more light on the collection of macroplastics on Wednesdays and Saturdays only since you said sampling was done for eight consecutive days?
Ø Line 119: Why a t-test? Did the samples meet the assumptions for a parametric test? If yes, state so. If no, use an alternative non-parametric test.
Lines 129-137: These do not belong to the methodology section. They should be under results.
Results
Ø Lines 167-170: Two contrasting values (2.88 items d-1 and 1.77 d-1) are presented as the average daily accumulation rate for rural beaches. Please, reconcile.
Ø Table 2 caption: Please write PPE in full if it’s being used for the first time.
Discussion
Ø I feel this section should be further enriched by discussing the implications of your findings on coastal ecosystems and human health. This study is very significant but it appears that the discussion is not rich enough. A plastic-laden beach has implications for human health, especially for eco-tourists who may find such an environment unattractive. Authors could find this and more other implications in the article “Plastic pollution threat in Africa: current status and implications for aquatic ecosystem health” useful for their discussion. They are also encouraged to cite more literature to underscore the implications.
Conclusion
Ø This is rather too long. Expunge lines 247-249 and write the whole section in one paragraph.
Ø Lines 259-262: There is need to grammar-check and rephrase the sentence.
Ø Line 264: Write as ‘environment-friendly’

Round 2
Reviewer 1 Report
The authors revised the manuscript significantly. However, there are still a few areas to improve.
1. Line 43-45: Microplastic size- Please express as a size range. Please see this reference FIGURE 11.1 Size-based classification of plastics. Microplastics in wastewater treatment plants (iges.or.jp)
Abeynayaka, A., Werellagama, I., Ngoc-Bao, P., Hengesbaugh, M., Gajanayake, P., Nallaperuma, B., Karkour, S., Bui, X.T. and Itsubo, N., 2022. Microplastics in wastewater treatment plants. Current Developments in Biotechnology and Bioengineering, pp.311-337.
2. Lines 52: Optional -"The Philippines is the biggest emitter of plastics from rivers to the marine environment [7]." Reference 7 mention this as a model prediction. Pelase revise this sentence to provide a more accurate meaning since this is the first line of the paragraph. i..e according to the model developed by ...
3. Figure 2.- Y axis units - items/m^2 or items/d.m^2 (wondering since in line 134 it is mentioned that items/d.m^2). Expressing the values in figure 2 is recommended in that unit (items/d.m^2).
4. Figure 3. Overall types of macroplastic litter in rural and urban beaches in Sarangani Bay, Philippines. Pelase consider providing example photographs of this item list as supplementary material. This will help global readers to understand the items easily. Also, this type of information is useful for machine learning kind of applications in plastic litter management (and improves the citability of this paper)
